# Quantum Discord, Thermal Discord, and Entropy Generation in the Minimum Error Discrimination Strategy

**DOI:** 10.3390/e21030263

**Published:** 2019-03-08

**Authors:** Omar Jiménez, Miguel Angel Solís-Prosser, Leonardo Neves, Aldo Delgado

**Affiliations:** 1Centro de Óptica e Información Cuántica, Facultad de Ciencias, Universidad Mayor, Camino La Pirámide N°5750, Huechuraba, Santiago, Chile; 2Departamento de Física, Universidad de Concepción, 160-C Concepción, Chile; 3Departamento de Física, Universidade Federal de Minas Gerais, Belo Horizonte-MG 31270-901, Brazil

**Keywords:** minimum error discrimination, accessible information, discord, second law of thermodynamics

## Abstract

We study the classical and quantum correlations in the minimum error discrimination (ME) of two non-orthogonal pure quantum states. In particular, we consider quantum discord, thermal discord and entropy generation. We show that ME allows one to reach the accessible information between the two involved parties, Alice and Bob, in the discrimination process. We determine the amount of quantum discord that is consumed in the ME and show that the entropy generation is, in general, higher than the thermal discord. However, in certain cases the entropy generation is very close to thermal discord, which indicates that, in these cases, the process generates the least possible entropy. Moreover, we also study the ME process as a thermodynamic cycle and we show that it is in agreement with the second law of thermodynamics. Finally, we study the relation between the accessible information and the optimum success probability in ME.

## 1. Introduction

Recently, quantum communication protocols have been studied from the point of view of the quantum correlations between the involved parties [1,2,3,4,5,6]. This allows one to quantify the resources that are required to carry out quantum communication protocols. Quantum correlations also allow us to differentiate between quantum and classical properties of a quantum state. The implementation of protocols for quantum communication requires that at least one of the parties implements a quantum measurement. This is, in general, an irreversible process [7] that changes the quantum state, produces decoherence and, also, entropy. Studied quantum correlations in bipartite scenarios are entanglement [8], quantum discord [9,10,11], thermal discord [12] and global discord [13,14].

The total amount of correlations contained in a bipartite state is quantified by the quantum mutual information, which represents the minimal amount of noise that is required to erase or destroy the total correlations in a many-copy scenario [15]. Quantum mutual information is also directly connected with Landauer’s original idea [16], which states that any logical irreversible process must dissipate entropy into the environment [17]. It can be cast as the sum of two terms: classical correlations and quantum discord. The latter, quantum discord, is defined in order to minimize the loss of quantum correlations due to quantum measurements [9]. Another measure of quantum correlation is the thermal discord. This takes into account the entropic cost of measurements and minimizes the entropy generation produced by quantum measurements [7,12]. Quantum discord and thermal discord are defined when one of the parties implements measurements in specific bases. These bases, in general, are different from each other. On the other hand, in some particular cases, such as in minimum error discrimination, we need to implement a measurement in a determined basis. For those cases, the basis-dependent versions of the aforementioned quantum correlations are considered [11]. In general, the discrimination of non-orthogonal states must satisfy the second law of thermodynamics [18,19] and, to demonstrate this, a central argument is the Landauer’s cost of erasure [12].

One of the most elemental quantum communication protocols is the discrimination among states belonging to a predefined set of known quantum states. If the set contains two or more non-orthogonal states, the discrimination of the quantum states cannot be carried out with certainty and deterministically [20]. In this case, we can resort to several discrimination strategies that optimize a predefined figure of merit. Minimum error discrimination (ME) is one of these strategies, where the discrimination of the non-orthogonal states is carried out in such a way that the probability of mistaking a retrodiction is minimized [21,22]. ME finds application in several quantum information processes such as quantum teleportation [23,24], entanglement swapping [25,26], quantum cryptography [27] and dense coding [28], among many others. The discrimination of non-orthogonal states by means of ME has been experimentally implemented for two states [29,30], for four states [31] in a two-dimensional Hilbert space, and for several sets of symmetric pure states in dimensions as high as 29 [32].

Here, we study the minimum error discrimination of two non-orthogonal states generated with arbitrary probabilities, in terms of the quantum correlations involved in the process. For this purpose, we consider the cases when Alice and Bob share a separable quantum channel. We determine the quantum discord and the thermal discord and compare these with the loss of correlations and the entropy generation in the case of ME. Moreover, we show that ME, when considered as a thermodynamic cycle, is in agreement with the second law of thermodynamics. Finally, we study the relation between the accessible information and the optimal probability of success in the ME protocol.

This article is organized as follows: In Section 2 we briefly review minimum error discrimination for two non-orthogonal states at the Helstrom limit. In Section 3 we describe the initial and final states of Alice and Bob after the measurement implemented by Bob according to ME. In Section 4 we study quantum and the thermal discord together with their relationship to the second law of thermodynamics. Moreover, the relation between the information gained by Bob and the optimum success probability in the ME, is studied. Finally, in Section 5 we summarize our results and conclude.

## 2. Minimum Error Discrimination

In the minimum error discrimination protocol, one of the communicating parties, let us say Bob, receives a single copy of a quantum system. This can be described by one of two possible non-orthogonal states in the set $\Omega ={\{|\phi_{0}\rangle }_{B},|\phi_{1}\rangle_{B}\}$. These states are generated with a priori probabilities $\eta_{0}$ and $\eta_{1}=1-\eta_{0}$, respectively. The set of states and a priori probabilities are known by Bob beforehand. Bob’s task is to identify, with the lowest average error, the probability of the state in Ω that describes the quantum system.

The states in Ω can be written as
(1a)$$\begin{array}{c} |\phi_{0}\rangle_{B}=\cos\frac{\beta }{2}{|0\rangle }_{B}+\sin\frac{\beta }{2}{|1\rangle }_{B}, \end{array}$$
(1b)$$\begin{array}{c} |\phi_{1}\rangle_{B}=\sin\frac{\beta }{2}{|0\rangle }_{B}+\cos\frac{\beta }{2}{|1\rangle }_{B}, \end{array}$$where the two orthonormal states ${\{|0\rangle }_{B}{,|1\rangle }_{B}\}$ form a base of the two-dimensional Hilbert space of Bob’s quantum system. The inner product between the non-orthogonal states is denoted by $\alpha =\langle \phi_{0}|\phi_{1}\rangle =\sin\beta$, where β∈[0,π/2], so that α∈[0,1].

In order to discriminate with ME between the non-orthogonal states ${\{|\phi_{0}\rangle }_{B},|\phi_{1}\rangle_{B}\}$, Bob first applies a unitary transformation $U_{B}$. In this case, $U_{B}$ can be written in the following form
(2a)$$\begin{array}{c} U_{B}|\phi_{0}\rangle_{B}=\sqrt{p_{0}}{|0\rangle }_{B}+\sqrt{r_{0}}{|1\rangle }_{B}, \end{array}$$
(2b)$$\begin{array}{c} U_{B}|\phi_{1}\rangle_{B}=\sqrt{r_{1}}{|0\rangle }_{B}+\sqrt{p_{1}}{|1\rangle }_{B}, \end{array}$$where, now, the orthonormal states ${\{|0\rangle }_{B}{,|1\rangle }_{B}\}$ represent the base in which Bob must implement his measurement in order to discriminate with ME probability. Here, $r_{i}$ ($p_{i}$) represents the probability of failure (success) in the identification of the state $|\phi_{i}\rangle_{B}$, where $p_{i}+r_{i}=1$. The unitarity of $U_{B}$ implies that the following constraint must be satisfied
(3)$$\alpha =\sqrt{r_{0}p_{1}}+\sqrt{r_{1}p_{0}}.$$

The average probability of error in the discrimination between the non-orthogonal states ${\{|\phi_{0}\rangle }_{B},|\phi_{1}\rangle_{B}\}$ is
(4)$$P_{e}=\eta_{0}r_{0}+\eta_{1}r_{1},$$where, $r_{i}=\left|\langle j\right|U_{B}|\phi_{i}{\rangle |}^{2}$ for i≠j. We reach the minimum average error probability in the discrimination process when the probabilities $r_{i}$ are given by [33]
(5)$$r_{i}=\frac{1}{2}\left(1-\frac{1-2\eta_{j}\alpha^{2}}{\sqrt{1-4\eta_{i}\eta_{j}\alpha^{2}}}\right),$$for i≠j. Therefore, the minimal average error probability attained by the minimum error discrimination strategy is given by
(6)$$P_{e}^{min}=\frac{1}{2}\left(1-\sqrt{1-4\eta_{0}\eta_{1}\alpha^{2}}\right),$$which is equal to the Helstrom limit [20,22]. The optimal average success probability in the discrimination is equal to $P_{s}^{opt}=1-P_{e}^{min}$. In what follows, given the symmetry with respect to the a priori probabilities, we consider the case $\eta_{1}\geq \eta_{0}$ for $0\leq \eta_{0}\leq 1/2$.

## 3. Channel without Entanglement

Let us consider initially that the communicating parties, Alice and Bob, share a separable quantum state of the form
(7)$$\rho_{AB}=\sum_{i=0}^{1}\eta_{i}{|i\rangle }_{A}\langle i|⊗|\phi_{i}\rangle_{B}\langle \phi_{i}|,$$where the states ${\{|0\rangle }_{A}{,|1\rangle }_{A}\}$ form an orthonormal base for Alice’s two-dimensional quantum system, and ${\{|\phi_{0}\rangle }_{B},|\phi_{1}\rangle_{B}\}$ are the two possible non-orthogonal states of Bob’s quantum system given by Equations (1). Alice prepares a single copy of a quantum system in the state $|\phi_{i}\rangle_{B}$ and sends it to Bob with a priori probability $\eta_{i}$. Thereby, Alice and Bob share quantum and classical correlations encoded in the joint state $\rho_{AB}$ of Equation (7). The initial state $\rho_{A}$ of Alice’s quantum system, that is, prior to the application of any transformation or measurement, is obtained by $\rho_{A}={\mathrm{tr}}_{B}\left(\rho_{AB}\right)$, where
(8)$$\begin{array}{ccc} \rho_{A} & = & \sum_{i=0}^{1}\eta_{i}{|i\rangle }_{A}\langle i|. \end{array}$$

In a similar form, the initial state of Bob’s quantum system can be obtained from $\rho_{B}={\mathrm{tr}}_{A}\left(\rho_{AB}\right)$, where
(9)$$\rho_{B}=\sum_{i=0}^{1}\eta_{i}|\phi_{i}\rangle_{B}\langle \phi_{i}|.$$

Once Bob has received the single copy of the quantum system in the state $|\phi_{i}\rangle_{B}$, he implements the optimal strategy of ME. For that purpose, Bob first applies the unitary transformation $U_{B}$ onto his quantum system. Thereby, the initial joint state $\rho_{AB}$ of Equation (7) changes to ${\hat{\rho }}_{AB}=\left(1_{A}⊗U_{B}\right)\rho_{AB}\left(1_{A}⊗U_{B}^{†}\right)$, where
(10)$${\hat{\rho }}_{AB}=\sum_{i=0}^{1}\eta_{i}{|i\rangle }_{A}\langle i|⊗|\hat{\phi_{i}}\rangle_{B}\langle \hat{\phi_{i}}|,$$with $|\hat{\phi_{i}}\rangle_{B}=U_{B}{|\phi_{i}\rangle }_{B}$. The unitary transformation $U_{B}$ of Equation (2), applied by Bob onto his quantum system, is a reversible process [7]. Therefore, it does not change the quantum correlations between Alice and Bob and it does not produce entropy either.

We consider that Bob can implement his measurement in an arbitrary basis ${\{|0^{\prime }\rangle }_{B},|1^{\prime }\rangle_{B}\}$, which is given by
(11a)$$\begin{array}{ccc} |0^{\prime }\rangle_{B} & = & {x|0\rangle }_{B}-y{|1\rangle }_{B}, \end{array}$$
(11b)$$\begin{array}{ccc} |1^{\prime }\rangle_{B} & = & {y|0\rangle }_{B}+x{|1\rangle }_{B}, \end{array}$$where the coefficients *x* and *y* are real positive numbers that satisfy $x^{2}+y^{2}=1$. The measurement carried out by Bob on his quantum system generates two conditional post-measurement states $\rho_{A|b}^{i}$ for Alice’s quantum system. Provided that Bob’s measurement projects his quantum system onto the state $|i^{\prime }\rangle_{B}$, Alice’s post-measurement states can be
(12)$$\begin{array}{c} \rho_{A|b}^{0}={(t_{00}^{2}|0\rangle }_{A}\langle 0|+t_{10}^{2}{|1\rangle }_{A}\langle 1|)/p_{0}^{b}, \end{array}$$
(13)$$\begin{array}{c} \rho_{A|b}^{1}={(t_{01}^{2}|0\rangle }_{A}\langle 0|+t_{11}^{2}{|1\rangle }_{A}\langle 1|)/p_{1}^{b}, \end{array}$$respectively, where
(14)$$\begin{array}{ccc} t_{00} & = & xt_{0}-ym_{0}, \end{array}$$
(15)$$\begin{array}{ccc} t_{01} & = & xm_{0}+yt_{0}, \end{array}$$
(16)$$\begin{array}{ccc} t_{10} & = & xm_{1}-yt_{1}, \end{array}$$
(17)$$\begin{array}{ccc} t_{11} & = & xt_{1}+ym_{1}, \end{array}$$with
(18)$$\begin{array}{ccc} m_{i} & = & \sqrt{\eta_{i}r_{i}}, \end{array}$$
(19)$$\begin{array}{ccc} t_{i} & = & \sqrt{\eta_{i}\left(1-r_{i}\right)}, \end{array}$$
(20)$$\begin{array}{ccc} p_{i}^{b} & = & t_{0i}^{2}+t_{1i}^{2}, \end{array}$$for i=0,1 and
(21)$$\sum_{i,j=0}^{1}t_{ij}^{2}=1.$$

The final average joint state between Alice and Bob $\rho_{AB}^{\prime }$, when Bob implements his measurement in the basis ${\{|0^{\prime }\rangle }_{B},|1^{\prime }\rangle_{B}\}$, takes the following form:(22)$$\rho_{AB}^{\prime }=\sum_{i=0}^{1}p_{i}^{b}\rho_{A|b}^{i}⊗\Pi_{b}^{i^{\prime }},$$where, $\Pi_{b}^{i^{\prime }}$ are the projectors $|i^{\prime }\rangle_{B}\langle i^{\prime }|$ onto the Hilbert space of Bob’s quantum system. The state $\rho_{AB}^{\prime }$ is a classical state because there are local measurements in Alice’s and Bob’s systems that do not perturb it [11]. The final reduced states for Alice’s and Bob’s quantum systems are given by
(23)$$\begin{array}{ccc} \rho_{A}^{\prime } & = & p_{0}^{b}\rho_{A|b}^{0}+p_{1}^{b}\rho_{A|b}^{1}, \end{array}$$
(24)$$\begin{array}{ccc} \rho_{B}^{\prime } & = & p_{0}^{b}|0^{\prime }\rangle_{B}\langle 0^{\prime }|+p_{1}^{b}|1^{\prime }\rangle_{B}\langle 1^{\prime }|. \end{array}$$

Thereby, the final reduced state for Alice’s system does not change, that is, $\rho_{A}^{\prime }=\eta_{0}{|0\rangle }_{A}\langle 0|+\eta_{1}{|1\rangle }_{A}\langle 1|=\rho_{A}$.

In the particular case of ME, that is, for x=1 in Equations (11), the average joint state $\rho_{AB}^{\prime }$ of Equation (22) takes the following form
(25)$$\rho_{AB}^{\prime }\left(x=1\right)=t_{0}^{2}{|00\rangle }_{AB}\langle 00|+m_{0}^{2}{|01\rangle }_{AB}\langle 01|+m_{1}^{2}{|10\rangle }_{AB}\langle 10|+t_{1}^{2}{|11\rangle }_{AB}\langle 11|.$$

In this particular case, if Bob successfully discriminates the state $|\phi_{i}\rangle_{B}$, the final state that Alice and Bob share will be ${|ii\rangle }_{AB}$. Otherwise, when the discrimination attempt is unsuccessful, the final state is |ij〉 with i≠j. The average error probability is equal to $p_{e}^{min}=m_{0}^{2}+m_{1}^{2}$ and agrees with the Helstrom bound given by Equation (6).

## 4. Correlations between Alice and Bob

### 4.1. Classical Correlations and Quantum Discord

In a bipartite state $\rho_{AB}$, the total amount of correlation, in the many copy scenario [15], is given by the quantum mutual information. This is defined as [9,15]
(26)$$I\left(\rho_{AB}\right)=S\left(\rho_{A}\right)+S\left(\rho_{B}\right)-S\left(\rho_{AB}\right),$$where S(ρ) is the von Neumann entropy of the state ρ, given by $S\left(\rho \right)=-\sum_{i}\lambda_{i}\log_{2}\lambda_{i}$, where $\lambda_{i}$ are the eigenvalues of ρ. Hence, in our scheme of ME discrimination, we consider that Alice emits many copies of independent identically distributed (i.i.d.) data, that is $\sigma =\rho_{AB}^{n}$ for some large *n* [34].

The quantum mutual information can be written as [9,11]
(27)$$I\left(\rho_{AB}\right)=J\left(A|\left\{\Pi_{b}\right\}\right)+D\left(A|\left\{\Pi_{b}\right\}\right),$$where, $J(A|\left\{\Pi_{b}\right\})$ are the classical correlations and $D(A|\left\{\Pi_{b}\right\})$ is the quantum discord. These two quantities depend on the measurement implemented by Bob, through the set of projectors $\left\{\Pi_{b}\right\}$, but their sum does not [2], i.e., they are complementary to each other. The classical correlations $J(A|\left\{\Pi_{b}\right\})$ between Alice and Bob are defined as [10,11]
(28)$$J\left(A|\left\{\Pi_{b}\right\}\right)=S\left(\rho_{A}\right)-\sum_{i=0}^{1}p_{i}^{b}S\left(\rho_{A|b}^{i}\right),$$which can be interpreted as the information about Alice’s system gained by Bob by means of the measurement $\left\{\Pi_{b}\right\}$. Here, we are interested in maximizing the classical correlation $J(A|\left\{\Pi_{b}\right\})$ with respect to all possible measurements implemented by Bob, that is
(29)$$J\left(A|B\right)=\max_{\left\{\Pi_{b}\right\}}J\left(A|\left\{\Pi_{b}\right\}\right)=S\left(\rho_{A}\right)-\min_{\left\{\Pi_{b}\right\}}\sum_{i=0}^{1}p_{i}^{b}S\left(\rho_{A|b}^{i}\right),$$which is called the accessible information [35,36]. This represents the classical mutual information maximized with respect to the detection strategy [31].

On the other hand, the minimum quantum discord, which quantifies the quantum correlations that are consumed or lost in the process, is given by
(30)$$D\left(A|B\right)=I\left(\rho_{AB}\right)-J\left(A|B\right).$$

To quantify the accessible information, we minimize the expression $f\left(x\right)=\sum_{i}p_{i}^{b}S\left(\rho_{A|b}^{i}\right)$, with respect to the variable *x* that defines the measurement base in Equation (11). The function f(x) can be cast as
(31)$$f\left(x\right)=-\sum_{i,j=0}^{1}t_{ij}^{2}\log_{2}\left(\frac{t_{ij}^{2}}{p_{j}^{b}}\right),$$and its derivative with respect to *x* is
(32)$$\begin{array}{c} \frac{d}{dx}f\left(x\right)=-\frac{2}{\sqrt{1-x^{2}}}\sum_{i=0}^{1}t_{i0}t_{i1}\log_{2}\left(\frac{t_{i0}^{2}p_{1}^{b}}{t_{i1}^{2}p_{0}^{b}}\right). \end{array}$$

This derivative vanishes at x=0. In fact, this a minimum. However, the function f(x) is such that f(x=0)=f(x=1) and consequently x=1 is also a minimum. In both cases the entropy of Alice’s conditional states are equal, $S\left(\rho_{A|b}^{0}\right)=S\left(\rho_{A|b}^{1}\right)$. In fact, the cases x=0 or x=1 are physically equivalent since they are connected by the transformations $|0^{\prime }\rangle \to |1^{\prime }\rangle$ and $|1^{\prime }\rangle \to |0^{\prime }\rangle$, which is equivalent to the exchange $|\phi_{0}\rangle_{B}\to {|\phi_{1}\rangle }_{B}$ and $|\phi_{1}\rangle_{B}\to {|\phi_{0}\rangle }_{B}$. On the other hand, to carry out ME discrimination, Bob must consider the basis of measurement with x=1 (or x=0) in Equations (11). Thereby, in these particular cases, the same measurement base reaches simultaneously ME discrimination and accessible information [37,38].

To determine the amount of quantum discord, we consider the expression
(33)$$D\left(A|B\right)=S\left(\rho_{B}\right)+\min_{\left\{\Pi_{b}\right\}}\sum_{i=0}^{1}p_{i}^{b}S\left(\rho_{A|b}^{i}\right)-S\left(\rho_{AB}\right),$$where the eigenvalues of Bob’s state $\rho_{B}$, given by Equation (9), are $\lambda_{0}^{b}=\left(1-\sqrt{1-4\eta_{0}\eta_{1}\cos^{2}\beta }\right)/2$ and $\lambda_{1}^{b}=1-\lambda_{0}^{b}$. Moreover, the entropy of the joint initial state $\rho_{AB}$, given by Equation (Equation 7), is $S\left(\rho_{AB}\right)=S\left(\rho_{A}\right)+\sum \eta_{i}S{(|\phi_{i}\rangle }_{B}\langle \phi_{i}|)=S\left(\rho_{A}\right)$.

Figure 1a,b show the accessible information and the quantum discord, respectively, for x=1, as a function of the inner product α for three decreasing values of $\eta_{0}$ (from top to bottom). The accessible information J(A|B) takes its maximum value, for a fixed value of $\eta_{0}$, when the states ${\{|\phi_{0}\rangle }_{B},|\phi_{1}\rangle_{B}\}$ are orthogonal (α=0) and it decreases with the inner product α, reaching its minimum value, J(A|B)=0, when α=1. Moreover, in the particular case $\eta_{0}=1/2$, Alice and Bob share the maximal mutual information available from any ensemble of quantum states [37]. If we consider the case $\eta_{0}=0$, the ME has only one state in the discrimination and the ME strategy does not convey any information at all. In this case we have that J(A|B)=0.

The quantum discord D(A|B) takes its minimum value equal to zero, for any value of $\eta_{0}$, when the states ${\{|\phi_{0}\rangle }_{B},|\phi_{1}\rangle_{B}\}$ are orthogonal (α=0) or their inner product is equal to one. In the aforementioned cases, Bob’s measurement does not change the joint state $\rho_{AB}^{\prime }={\hat{\rho }}_{AB}$ and therefore, the states ${\{|\phi_{0}\rangle }_{B},|\phi_{1}\rangle_{B}\}$ behave like classical states. On the other hand, for a fixed value of α the quantum discord takes its maximal value when $\eta_{0}=0.5$. Simultaneously, the accessible information for Bob is maximal in this case. As is apparent from Figure 1b, the maximum of the quantum discord occurs at $\alpha =1/\sqrt{2}$ for any value of $\eta_{0}$. This happens because D(A|B) is a concave function of $\alpha^{2}$ and it is symmetric under interchange of $\alpha^{2}$ and $1-\alpha^{2}$. Physically, we can say that when $\alpha^{2}=1/2$, the states ${\{|\phi_{0}\rangle }_{B},|\phi_{1}\rangle_{B}\}$ are in an intermediate position between the two cases $\alpha^{2}=0$ and $\alpha^{2}=1$, where the quantum discord vanishes and, consequently, there are no quantum correlations. On the other hand, the case $\alpha^{2}=1/2$ is the least similar to the aforementioned ones and, therefore, a maximum value of the quantum discord is to be expected.

### 4.2. Thermal Discord

Another measure of quantum correlations that we consider in the analysis of ME discrimination, is the thermal discord. This takes into account the entropic cost of realizing local measurements. The thermal discord is defined by [11,12]
(34)$$D_{th}\left(A|B\right)=\min_{\left\{\Pi_{b}\right\}}\left[S\left(\rho_{B}^{\prime }\right)+\sum_{i=0}^{1}p_{i}^{b}S\left(\rho_{A|b}^{i}\right)\right]-S\left(\rho_{AB}\right).$$

In this case, the term $S\left(\rho_{B}^{\prime }\right)+\sum_{i=0}^{1}p_{i}^{b}S\left(\rho_{A|b}^{i}\right)$ to be minimized in the Equation (Equation 34) turns out to be equal to the entropy $S(\rho_{AB}^{\prime })$ of the state $\rho_{AB}^{\prime }$ of Equation (Equation 22), which corresponds to the joint state after Bob’s measurement. Hence, the thermal discord is equal to the minimum entropy generation due to the measurement implemented by Bob, which is also equal to the one-way quantum deficit [7,11,39]. Then the measurement-dependent thermal discord (DTD) is
(35)$$D_{th}\left(A|\left\{\Pi_{b}\right\}\right)=S\left(\rho_{AB}^{\prime }\right)-S\left(\rho_{AB}\right),$$and it corresponds to entropy generation due to Bob’s measurement in a particular basis $\left\{\Pi_{b}\right\}$.

In general, the thermal discord is higher than or equal to the quantum discord [7,12,40], that is,
(36)$$D_{th}\left(A|B\right)\geq D\left(A|B\right).$$

This indicates that the minimal generation of entropy must be higher or equal to the minimum of the quantum correlations that are destroyed due to Bob’s measurement [16]. This inequality is related to the second law of thermodynamics [12], by means of the entropic cost to take Bob’s system to the initial state. Given that a projective measurement does not decrease the entropy [41], we have $S\left(\rho_{B}^{\prime }\right)\geq S\left(\rho_{B}\right)$ and this is valid for any base chosen by Bob for implementing his measurement. On the other hand, from the condition $I\left(\rho_{AB}\right)\geq J\left(A|B\right)$ we obtain that $S\left(\rho_{B}\right)\geq J\left(A|B\right)$. Thus, we have that
(37)$$J\left(A|B\right)\leq S\left(\rho_{B}^{\prime }\right).$$

Equation (Equation 37) can be understood if we consider the scheme of ME as a thermodynamic cycle [17,42]. The second law establishes that the net work after completing a cycle cannot be positive, that is $W_{out}-W_{in}\leq 0$. The work $W_{out}$ that can be extracted is proportional to the classical mutual information J(A|B) between Alice and Bob. The work invested $W_{in}$ is proportional to the Holevo bound $S\left(\rho_{B}\right)-\sum \eta_{i}S{(|\phi_{i}\rangle }_{B}\langle \phi_{i}|)=S\left(\rho_{B}\right)$ plus the minimal erasure work to take the final state $\rho_{B}^{\prime }$ to the initial state $\rho_{B}$, which is equal to $S\left(\rho_{B}^{\prime }\right)-S\left(\rho_{B}\right)\geq 0$. From this, it is clear that $W_{in}=k_{B}TS\left(\rho_{B}^{\prime }\right)$ and $W_{out}=k_{B}TJ\left(A|B\right)$, where $k_{B}$ is the Boltzmann constant and *T* is the temperature of the thermal reservoir. Thereby, the second law of thermodynamics is satisfied [17,18,19].

In our case, to determine the thermal discord and the entropy generation in Bob’s measurement, we consider that $S\left(\rho_{B}^{\prime }\right)=-p_{0}^{b}\log_{2}p_{0}^{b}-p_{1}^{b}\log_{2}p_{1}^{b}$. For the entropy generation in ME, we evaluate the function of Equation (Equation 35) for x=1. On the other hand, the DTD in Equation (35) is a function of the variable *x* that defines the measurement base in Equations (11) and it can be cast as
(38)$$D_{th}\left(A|\left\{\Pi_{b}\right\}\right)\left(x\right)=-\sum_{i,j=0}^{1}t_{ij}^{2}\log_{2}\left(t_{ij}^{2}\right)-S\left(\rho_{A}\right).$$

In order to find the thermal discord, we need to minimize it with respect to the variable *x*, then we consider the restriction $\frac{d}{dx}D_{th}\left(A|\left\{\Pi_{b}\right\}\right)\left(x\right)=0$. The last condition takes the following form
(39)$$\frac{1}{\sqrt{1-x^{2}}}\left(t_{00}t_{01}\log_{2}\left(\frac{t_{00}^{2}}{t_{01}^{2}}\right)+t_{10}t_{11}\log_{2}\left(\frac{t_{10}^{2}}{t_{11}^{2}}\right)\right)=0.$$

We solve Equation (Equation 39) numerically in order to find the values of *x* where the function $D_{th}\left(A|\left\{\pi_{b}\right\}\right)\left(x\right)$ takes its minimum values.

Figure 2a displays the values of *x* that allow us to attain the minimal value of the measurement-dependent thermal discord, that is, the thermal discord as a function of the inner product α for three values of $\eta_{0}$. For $\eta_{0}=0.05$ and $\eta_{0}=0.2$ the variable *x* is close to the unity in the complete interval of α values. For η≈0.5 the values of *x* depart from the unity when α is approximately equal to or larger than 0.5.

In the interval α∈[0,0.5] the minimum error probability increases very slowly and is upper bounded by the value 0.07 for any value of $\eta_{0}$. Therefore, the discrimination process achieves a high accuracy. In this regime, the coefficients $m_{i}^{2}$ are very small. As can be seen from Equation (Equation 5), the $m_{i}^{2}$ coefficients are upper bounded by the value 0.035. In this case, the post-measurement state for ME is approximately given by $\rho_{AB}^{\prime }\approx t_{0}^{2}{|00\rangle }_{AB}\langle 00|+t_{1}^{2}{|11\rangle }_{AB}\langle 11|$. A similar state can also be obtained from Equation (22) considering a first order Taylor series expansion for $m_{i}$ and $\sqrt{1-x^{2}}$. The latter is suggested because it is easy to show that the base-dependent thermal discord is optimized at x=1 for $\alpha^{2}=0$. Thus, for small values of $\alpha^{2}$ the Taylor series expansion is a good guess. The state obtained in this way has four eigenvalues that are functions of *x*. In order to minimize the entropy of this state, we can vanish two of the eigenvalues by choosing x=1 independently of the value of $\eta_{0}$. This procedure leads to a state that agrees with the state $\rho_{AB}^{\prime }$ above. Therefore, when the minimum error probability $P_{e}^{min}$ is very small, that is, for α≤0.5, entropy generation in the ME process approximately agrees with the value of the thermal discord.

Figure 2b shows the generation of entropy in ME in solid lines and the thermal discord in dashed lines, as a function of the inner product α and for three values of $\eta_{0}$. We obtained that the entropy generation in ME is higher than or equal to the thermal discord. This means that, in general, the base for the thermal discord and for ME do not coincide. However, for small values of α the entropy generation is very close to the thermal discord and consequently, in these cases, ME is also almost optimal from the point of view of the entropy generation.

When the states ${\{|\phi_{0}\rangle }_{B},|\phi_{1}\rangle_{B}\}$ are orthogonal (α=0) the entropy generation and thermal discord are zero because in this case, as already pointed out, the joint state does not change, that is, $\rho_{AB}^{\prime }=\rho_{AB}$. On the other hand, in the case α=1 and $\eta_{0}=0$, the state $\rho_{AB}$ is a product state and thus the entropy generation and thermal discord are also zero. The maximum value of entropy generation in ME occurs for $\eta_{0}=0.5$ as in the case of the maximum accessible information J(A|B) between Alice and Bob. As in the case of quantum discord, thermal discord is a concave function of $\alpha^{2}$ and also has its maximum when $\alpha^{2}=1/2$ for any value of $\eta_{0}$. This means that quantum correlations (quantum discord and thermal discord) indicate that the biggest disturbance, due to the quantum measurement, of the initial joint state $\rho_{AB}$ will be produced if $\alpha^{2}=1/2$. However, in general, the process of ME generates more entropy than thermal discord as we see in Figure 2b. The maximum of the entropy generation now arises at values closer to $\alpha^{2}=1$, depending on the value of $\eta_{0}$. We see that for large overlaps (which means a more difficult discrimination process) the entropy generation departs from the thermal discord. In addition, as $\eta_{0}$ increases, this departure becomes more “dramatic”. Large values of $\eta_{0}$ implies in less bias towards some state in the set. So, large overlaps and large $\eta_{0}$ bring more difficulty in the discrimination process and require a larger effort to perform ME. The extreme case would be $\eta_{0}=0.5$ and α=1. Finally, we would like to point out that the choice of $\eta_{0}=0.49$ was considered in order to show that when α=1 the entropy generation goes to zero.

Figure 3a compares the average error probability of ME (solid lines) to the average error probability obtained when employing the base that optimizes the thermal discord (dashed lines). As is apparent from the figure, the base associated with the thermal discord discriminates the states ${\{|\phi_{0}\rangle }_{B},|\phi_{1}\rangle_{B}\}$ almost as good as ME in the interval α∈[0,0.5] for the three inspected values of $\eta_{0}$. For α equal or larger than 0.5, ME delivers the smallest average error probability. Thus, the base that optimizes the measurement-dependent thermal discord provides a discrimination almost as good as optimal ME in the interval α∈[0,0.5] for the inspected values of $\eta_{0}$.

### 4.3. Accessible Information and Optimum Success Probability

Finally, we study the relation between the accessible information J(A|B) and the optimal average success probability in the minimum error discrimination. This can be done because the bases for accessible information and optimal ME coincide. Figure 3b shows that for a fixed value of $\eta_{0}$, the accessible information J(A|B) increases as a function of the optimum probability of discrimination $P_{s}^{opt}$. When the states are orthogonal, i.e., α=0, then $P_{s}^{opt}=1$ and the accessible information is maximal for a fixed value of $\eta_{0}$. On the other hand, if the states are equal, i.e., α=1, then $P_{s}^{opt}=\eta_{1}$ and the accessible information is equal to zero.

## 5. Conclusions

We studied ME discrimination for two non-orthogonal states generated with arbitrary a priori probabilities from the point of view of the quantum correlations involved in the process. We recovered a previously known result, namely, optimal ME allows us also to attain the accessible information between the communicating parties, Alice and Bob. Thereby, in the ME discrimination process it is possible to optimize simultaneously the average success probability as well as the information gained by Bob through the measurement.

In general, the base that optimizes the measurement-dependent thermal discord does not agree with the base that leads to ME. This implies that ME generates more entropy than the minimum possible given by the thermal discord. However, for values of the inner product α in the interval [0,0.5], the entropy generated in the ME process is very close to the one obtained in the thermal discord. This indicates that for these cases, the ME process is also efficient in terms of the generation of entropy. Furthermore, when the discrimination process is carried out by measuring onto the base that leads to the thermal discord, the average error probability becomes very close to the optimal value when α∈[0,0.5].

Quantum discord and thermal discord are zero when the states are orthogonal (α=0) or when we have only one state α=1 or $\eta_{0}=0$. Hence, in these cases, the ME protocol presents only a classical behavior, i.e., the initial state does not change and there is no entropy generation with Bob’s measurement. Otherwise, the scheme of ME presents quantum properties given that the quantum discord and the thermal discord are greater than zero. Moreover, the quantum discord maximum and the thermal discord maximum occur when $\alpha^{2}=1/2$, which is the intermediate point between the two classical cases α=0 and α=1.

We showed that the process of ME discrimination satisfies the second law of thermodynamics when it is considered as a thermodynamic cycle. Finally, we obtained that the amount of accessible information increases as a function of the optimal average success probability of discrimination in the minimum error discrimination strategy.

Here, we have studied the case of ME for two pure states. ME can also be formulated for an arbitrary finite number of states, which points out to the possibility of generalizing our results to this scenario. However, analytical solutions for ME are known in very few special cases such as, for instance, sets of equally likely pure symmetric states [43] or two states with arbitrary prior probabilities. Furthermore, in certain situations, when the number of states is larger than the dimension of the underlying Hilbert space, the solution of ME requires the use of a positive-operator valued measure, which is not analytically known. Therefore, the lack of analytical solutions prevents us from extrapolating our results to more complex scenarios for ME. This also applies to other optimal discrimination strategies, being these also complex optimization problems.

A feasible extension of our results might appear in the so-called sequential discrimination [44]. In this scenario, several parties attempt to discriminate among a set of states in such away that each one of them has access to the post-measurement states generated by the other parties. All parties cannot resort to classical communications. The strategy optimizes the joint discrimination probability. Here, we could study the change of the classical and quantum correlations as the quantum system encoding the unknown states passes from party to party.

## Figures and Tables

**Figure 1 entropy-21-00263-f001:**
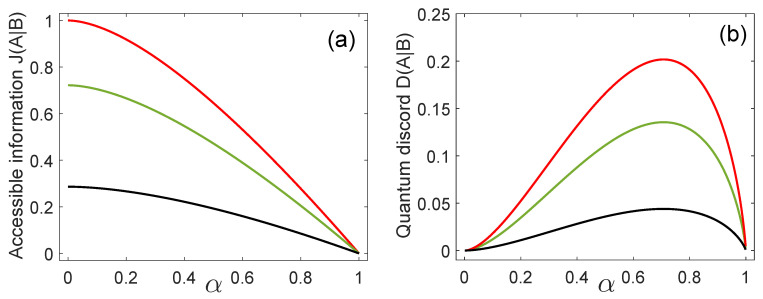
(**a**) Accessible information and (**b**) quantum discord, when x=1, as a function of the inner product α for $\eta_{0}=0.5$ (red line), $\eta_{0}=0.2$ (green line), and $\eta_{0}=0.05$ (black line).

**Figure 2 entropy-21-00263-f002:**
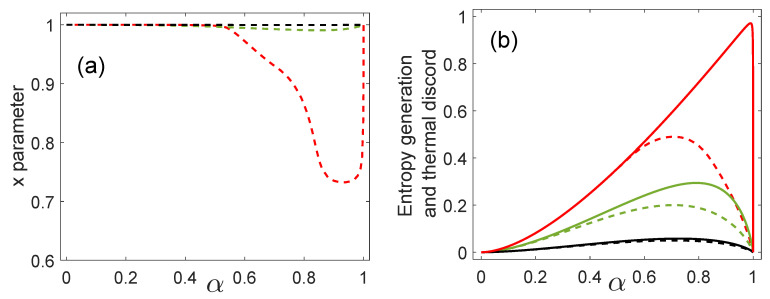
(**a**) Values of *x* that attain the thermal discord. (**b**) Entropy generation in minimum error (ME) (solid line) and thermal discord (dashed line) as a function of the inner product α for: $\eta_{0}=0.49$ (red line), $\eta_{0}=0.2$ (green line), and $\eta_{0}=0.05$ (black line).

**Figure 3 entropy-21-00263-f003:**
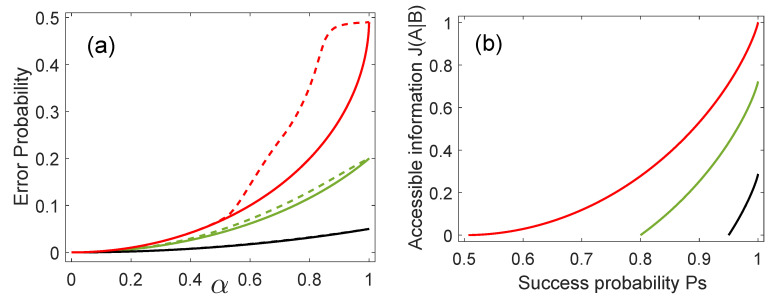
(**a**) Average error probability in ME (solid lines) and for the case when the measurement base leads to the thermal discord (dashed lines). (**b**) Accessible information versus average success probability in ME for $\eta_{0}=0.49$ (red line), $\eta_{0}=0.2$ (green line), and $\eta_{0}=0.05$ (black line).

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
