# Peer review of "Quantum Discord, Thermal Discord, and Entropy Generation in the Minimum Error Discrimination Strategy"

_entropy, 2019, doi:10.3390/e21030263_

Round 1
Reviewer 1 Report
In this work, the authors investigate the quantum discord and thermal discord in the protocol of minimal error discrimination.
After recalling in a clear way this protocol, they first consider the case where no quantum correlations are present between Alice and Bob. Then, they go to the case where quantum correlations are present. An important issue is to derive the expressions of the different quantities as a function of the measurement (its basis) performed by Bob. They nicely show that the quantum discord takes its maximal when the maximal amount of information is available for Bob. This conclusion is not valid in general for the thermal discord. Indeed, it only holds when the overlap between the two initial non-orthogonal states is between 0 and 0.5. Otherwise, the thermal discord is in general higher than the quantum discord.
To my opinion, this is an interesting work, clearly and well written, that contributes to the general and important question about the link between quantum information and quantum thermodynamics. Here, the authors take a well-known protocol from quantum information theory, and compare the quantities from quantum information theory and inspired by thermodynamics.
Personally, I would be interested in understanding in more details how the authors understand the good matching between quantum discord and thermal discord for \alpha \in [0, 0.5] and the bad matching for other values. Is there a fundamental reason behind? Is it possible to answer this question right now? If not, do the authors have ideas to investigate it?
Apart from that, I would recommend this work to be published in its current form. I found some typos that are listed below. Let me say that I did not check Eqs. 12-20. They seem to be reasonable. The authors could maybe provide in an appendix the details of these calculations, such that one can directly follow all steps.
With my best regards.
Typos:
l. 16: also allow us TO differentiate
l. 18: for quantum COMMUNICATION
l. 134: BEHAVE
l.144: From this, IT is clear
l. 165: AS apparent
Author Response
Response to Reviewer 1 Comments
We thank Reviewer 1 for having revised our manuscript.
Personally, I would be interested in understanding in more details how the authors understand the good matching between quantum discord and thermal discord for \alpha \in [0, 0.5] and the bad matching for other values. Is there a fundamental reason behind? Is it possible to answer this question right now? If not, do the authors have ideas to investigate it?
Response: In the interval α∈[0,0.5] the minimum error probability increases very slowly and is upper bounded by the value 0.07 for any value of η0. Therefore, the discrimination process achieves a high accuracy. In this regime, the coefficients mi2 are very small (mi2<0.035). In this case, the post-measurement state for ME is approximately given by ρAB' ≈ t02|00>AB<00|+t12|11>AB<11|. A similar state can also be obtained from Eq. (22) considering a first order Taylor series expansion for mi and \sqrt{1-x^2}. The latter is suggested because it is easy to show that the base-dependent thermal discord is optimized at x=1 for α2=0. Thus, for small values of α2 the Taylor series expansion is a good guess. The state obtained in this way has four eigenvalues that are functions of x. In order to minimize the entropy of this state, we can vanish two of the eigenvalues by choosing x=1 independently of the value of η0. This procedure leads to a state that agrees with the state ρAB' above. Therefore, when the minimum error probability Pemin is very small, that is, for α≤0.5, entropy generation in the ME process approximately agrees with the value of the thermal discord.
Apart from that, I would recommend this work to be published in its current form. I found some typos that are listed below. Let me say that I did not check Eqs. 12-20. They seem to be reasonable. The authors could maybe provide in an appendix the details of these calculations, such that one can directly follow all steps.
Response: The Eqs. 14-20 are only definitions for the coefficients that appear in Eqs. 12-13. Therefore, we think that in this part, it is not necessary to include an appendix with the calculations in the manuscript.
Typos:
l. 16: also allow us TO differentiate
l. 18: for quantum COMMUNICATION
l. 134: BEHAVE
l.144: From this, IT is clear
l. 165: AS apparent
Response: We included all the suggestions changes for the typos in the manuscript.
Reviewer 2 Report
In this manuscript, the minimum error discrimination of two non-orthogonal states generated with arbitrary probabilities is studied with regard to the quantum correlations involved in the process. The studied quantities are the accessible information, quantum discord and the thermal discord.
The manuscript is well-written and well-presented. The following comments are for the authors consideration.
In the discussion of the quantum discord in Fig. 1(b), there is no interpretation of the maxima whose positions seem to be independent of the chosen value of eta_0. Do they occur at alpha=1/sqrt(2)?
Is there any reason why eta_0=0.49 is used in Fig. 2 while eta_0=0.5 is used in Fig. 1?
What is the interpretation of the minimum for eta_0=0.49 in Fig. 2(a)?
What is the interpretation of the maxima in the entropy generation, thermal discord observed in Fig. 2(b)? Is there a reason why the position of the maxima in the thermal discord is shifted to smaller alpha, as compared to the ones in the entropy generation?
lines 176-177. I has trouble understanding this since it seems that it does not pertain to what is presented in Fig. 3(b). This seems to be the same as that stated on lines 126-127. In line 129, does eta_0=0 imply that alpha=1? Perhaps this could be made clearer in the Introduction.
line 9. ‘thermodynamics’. line 92. ‘Alice’s’.
Author Response
Response to Reviewer 2 Comments
We thank Reviewer 2 for having revised our manuscript.
In the discussion of the quantum discord in Fig. 1(b), there is no interpretation of the maxima whose positions seem to be independent of the chosen value of eta_0. Do they occur at alpha=1/sqrt(2)?
Response: Yes, they do, for any value of eta_0 the maximum of quantum discord is when α2=1/2. This happens because D(A|B) is a concave function of α2 and it is symmetric under interchange of α2 and 1-α2. Physically, we can say that when α2=1/2, the states {|ɸ0>B, |ɸ1>B} are in an intermediate position between the two cases α2=0 and α2=1, where the quantum discord vanishes and, consequently, there are no quantum correlations. On the other hand, the case α2=1/2 is the least similar to the aforementioned ones and, therefore, a maximum value of the quantum discord is to be expected.
Is there any reason why eta_0=0.49 is used in Fig. 2 while eta_0=0.5 is used in Fig. 1?
Response: The choice of η0=0.49 was considered in order to show that the entropy generation is a continuous function and that it goes to zero when α=1.
What is the interpretation of the minimum for eta_0=0.49 in Fig. 2(a)?
Response: There is no interpretation at the moment.
What is the interpretation of the maxima in the entropy generation, thermal discord observed in Fig. 2(b)? Is there a reason why the position of the maxima in the thermal discord is shifted to smaller alpha, as compared to the ones in the entropy generation?
Response: Thermal discord is a concave function of α2 and also it has its maximum when α=1/\sqrt(2) for any value of η0. On the other hand, the process of ME generates more entropy than the thermal discord as we can see in Fig.2b. The maximum for the entropy generation now approaches α=1, depending on the value of η0.
Lines 176-177. I has trouble understanding this since it seems that it does not pertain to what is presented in Fig. 3(b). This seems to be the same as that stated on lines 126-127. In line 129, does eta_0=0 imply that alpha=1? Perhaps this could be made clearer in the Introduction.
Response: The case η0 does not imply α=1. This can be seen from Eq.(6) for the minimum error probability. If we consider η0=0, this implies that Pemin=0. On the other hand, if we take α=1 then, we have a different expression, Pemin=(1-4*η0*η1)/2.
In the manuscript, we replaced the sentence: “When the states are equal, that is for α=1, Bob cannot obtain any classical information by his measurement and J(A|B)=0.” by “When the states are orthogonal, i.e., α=0, then Psopt=1 and the accessible information is maximal for a fixed value of η0. On the other hand, if the states are equal, i.e., α=1, then Psopt= η1 and the accessible information is equal to zero.”
Line 9. ‘thermodynamics’. line 92. ‘Alice’s’.
Response: We included the suggestions changes for the typos in the manuscript.
Reviewer 3 Report
The authors present an analysis of the entropic resources consumed in a state discrimination protocol. In particular they consider a specific class of bipartite, qubit states that are separable. They then apply the minimal error (ME) discrimination protocol and examine the quantum discord, thermal discord, and entropy generated. They show that within a particular regime the thermal discord and entropy generation are close to one-another and as such conclude that this process is close to thermodynamically optimal.
The article is generally well written and coherent in its presentation. I suspect that the introductory sections would be particularly useful for a pedagogical exposition of the protocol. I have a number of comments/queries mainly related to the generality and novelty of the current manuscript that I would ask the authors to consider before I feel I can recommend publication in Entropy.
1a) The setting is a simply two-qubit problem and as far as I can tell the main contribution of the authors here is to compute the various figures of merit of the two-qubit state. This makes the novelty of the article somewhat low in my opinion. Of course, such settings are often ideal for exemplifying a particular result. However, here I am not convinced that the conclusions drawn by the authors extend to more complex settings, possibly multipartite or higher dimensional. I suggest the authors at least consider including some discussions on the general validity of their conclusions - can one expect similar behaviours in settings other than a simple 2 qubit setting.
1b) A closely related point, the authors consider a special class of states as given by Eq. (7). Just as Landauer’s principle can be violated when initial correlations are shared, would the authors expect that their results would breakdown if the initial state shared some strong correlations possibly in the form of entanglement? Is is possible to examine the same setting but instead considering the Werner states as initial states - thus allowing one to consider both separable and entangled initial states?
2) I found the exposition and discussions of the results rather descriptive. The authors largely state the behaviour without attempting to provide any physical insight into. For example, why for \alpha \in [0,0.5] the thermal and entropy generation are almost equal? Indeed from Fig. 2a we see \alpha>0.5 leads to a significant change in the ‘x’ parameter, but again the authors fail to provide any physical insight into the underlying mechanisms. I feel this is something the article is lacking throughout.
3) At variance with the above features, we see the discord and thermal discord are maximized at \alpha=1/\sqrt{2}. What is special about this value, mathematically? physically?
In summary, the present article may warrant publication in Entropy however I feel some significant modifications are in order before this referee would recommend the article be accepted.
Author Response
Response to Reviewer 3 Comments
We thank Reviewer 3 for having revised our manuscript.
1a) The setting is a simply two-qubit problem and as far as I can tell the main contribution of the authors here is to compute the various figures of merit of the two-qubit state. This makes the novelty of the article somewhat low in my opinion. Of course, such settings are often ideal for exemplifying a particular result. However, here I am not convinced that the conclusions drawn by the authors extend to more complex settings, possibly multipartite or higher dimensional. I suggest the authors at least consider including some discussions on the general validity of their conclusions - can one expect similar behaviours in settings other than a simple 2 qubit setting.
Response 1a: We added the following paragraphs in the conclusions, aimed to address this concern:
Here, we have studied the case of ME for two pure states. ME can also be formulated for an arbitrary finite number of states, which points out to the possibility of generalizing our results to this scenario. However, analytical solutions for ME are known in very few special cases such as, for instance, sets of equally likely pure symmetric states [Ban et al, IJTP 36, 1269 (1997)] or two states with arbitrary prior probabilities. Furthermore, in certain situations, when the number of states is larger than the dimension of the underlying Hilbert space, the solution of ME requires the use of a positive-operator valued measure, which is not analytically known. Therefore, the lack of analytical solutions prevents us from extrapolating our results to more complex scenarios for ME. This also applies to other optimal discrimination strategies, being these also complex optimization problems.
A feasible extension of our results might appear in the so-called sequential discrimination [Bergou et al, PRL 111, 100501 (2013)]. In this scenario, several parties attempt to discriminate among a set of states in such away that each one of them has access to the post-measurement states generated by the other parties. All parties cannot resort to classical communications. The strategy optimizes the joint discrimination probability. Here, we could study the change of the classical and quantum correlations as the quantum system encoding the unknown states passes from party to party.
1b) A closely related point, the authors consider a special class of states as given by Eq. (7). Just as Landauer’s principle can be violated when initial correlations are shared, would the authors expect that their results would breakdown if the initial state shared some strong correlations possibly in the form of entanglement? Is is possible to examine the same setting but instead considering the Werner states as initial states - thus allowing one to consider both separable and entangled initial states?
Response 1b: If initially Alice and Bob share an entangled state, and Alice and Bob can do both measurements, we need to use other types of quantum correlations such as the global discord or the measurement-induced disturbance. In this case, we think that we can have different and unexpected results. This could be interesting to investigate in the near future.
Moreover, if we consider a Werner state, for instance, the quantum discord does not depends on the measurement basis but it only depends on the parameter “z” that defines the Werner state, see PRL 88, 017901 (2001). The Werner state has the following form
ρ = z|φ><φ|+(1-z)1/N
And a joint state between Alice and Bob that allows us to implement ME with entanglement has the following form
|Φ>AB = sqrt{η0}|0>A|φ0>B + sqrt{η1}|1>A|φ1>B,
which we need to use in order to study the quantum correlations present in the ME in a scheme with entanglement.
2) I found the exposition and discussions of the results rather descriptive. The authors largely state the behaviour without attempting to provide any physical insight into. For example, why for \alpha \in [0,0.5] the thermal and entropy generation are almost equal? Indeed from Fig. 2a we see \alpha>0.5 leads to a significant change in the ‘x’ parameter, but again the authors fail to provide any physical insight into the underlying mechanisms. I feel this is something the article is lacking throughout.
Response 2: We added the following sentence to the manuscript (lines 162-174):
In the interval α∈[0,0.5] the minimum error probability increases very slowly and is upper bounded by the value 0.07 for any value of η0. Therefore, the discrimination process achieves a high accuracy. In this regime, the coefficients mi2 are very small. As can be seen from Eq. (5), the mi2 coefficients are upper bounded by the value 0.035. In this case, the post-measurement state for ME is approximately given by ρAB' ≈ t02|00>AB<00|+t12|11>AB<11|. A similar state can also be obtained from Eq. (22) considering a first order Taylor series expansion for mi and \sqrt{1-x^2}. The latter is suggested because it is easy to show that the base-dependent thermal discord is optimized at x=1 for α2=0. Thus, for small values of α2 the Taylor series expansion is a good guess. The state obtained in this way has four eigenvalues that are functions of x. In order to minimize the entropy of this state, we can vanish two of the eigenvalues by choosing x=1 independently of the value of η0. This procedure leads to a state that agrees with the state ρAB' above. Therefore, when the minimum error probability Pemin is very small, that is, for α≤0.5, entropy generation in the ME process approximately agrees with the value of the thermal discord.
3) At variance with the above features, we see the discord and thermal discord are maximized at \alpha=1/\sqrt{2}. What is special about this value, mathematically? physically?
Response 3: As in the case of quantum discord, thermal discord is a concave function of α2 and they are symmetric under exchange of α2 by 1-α2. Therefore, the quantum discord maximum and thermal discord maximum occur when α=1/\sqrt{2} for any value of η0. Physically, in the case α2=1/2, the non-orthogonal states are in an intermediate position between the two classical cases α=0 and α=1. This means that the quantum correlations (quantum discord and thermal discord) indicate that the biggest disturbance in the initial joint state ρAB will be produced if α=1/\sqrt{2}. However, in general, the process of ME generates more entropy than the thermal discord as we see in Fig. 2b. The maximum for the entropy generation now approaches α=1, depending on the value of η0.
Round 2
Reviewer 3 Report
The authors have largely addressed my concerns so I would support publication in Entropy.